# Assessing the Impact of External and Internal Factors on Emergency Department Overcrowding

**DOI:** 10.3390/healthcare13202577

**Published:** 2025-10-14

**Authors:** Abdulaziz Ahmed, Khalid Y. Aram, Mohammed Alzeen, Orhun Vural, James Booth, Brittany F. Lindsey, Bunyamin Ozaydin

**Affiliations:** 1Department of Health Services Administration, School of Health Professions, University of Alabama at Birmingham, Birmingham, AL 35233, USA; 2Department of Biomedical Informatics and Data Science, Heersink School of Medicine, University of Alabama at Birmingham, Birmingham, AL 35233, USA; 3School of Business & Technology, Emporia State University, Emporia, KS 66801, USA; 4Department of Electrical and Computer Engineering, University of Alabama at Birmingham, Birmingham, AL 35233, USA; 5Department of Emergency Medicine, University of Alabama at Birmingham, Birmingham, AL 35233, USA; 6Department of Patient Throughput, University of Alabama at Birmingham, Birmingham, AL 35233, USA

**Keywords:** emergency department, emergency overcrowding, waiting count, weather and emergency visits, football game and emergency visits, linear regression, quantitative analysis

## Abstract

**Simple Summary:**

Emergency departments often have long wait times, but we do not fully understand all the factors that contribute to ED overcrowding. This study looked at how weather, football games, holidays, and hospital operations affect ED waiting times at a major medical center over four years. We found that bad weather, especially thunderstorms, leads to more people waiting in the ED. Surprisingly, clear weather also increased wait times. Football games caused more crowding 12 h before game time, likely because of pre-game injuries and celebrations. Weekends and federal holidays had fewer people waiting, probably because people delay non-urgent visits when regular doctors are not available. The number of patients stuck in the ED waiting for hospital beds (boarding) and overall hospital fullness showed complex patterns. When measured at different time points, their effects changed from increasing to decreasing wait times, showing that timing matters in understanding ED crowding. These findings can help hospitals better predict busy periods and adjust staffing. For example, hospitals could add extra staff before thunderstorms or football games. Understanding these patterns helps hospitals prepare for crowding before it happens, potentially reducing your wait time when you need emergency care.

**Abstract:**

**Objectives**: This study analyzes factors influencing Emergency Department (ED) overcrowding by examining the impacts of operational, environmental, and external variables, including weather conditions and football games. **Materials and Methods**: This retrospective observational study analyzed emergency department (ED) tracking and hospital census data from a southeastern U.S. academic medical center covering 2019–2023. These data were merged with corresponding weather, football event, and federal holiday data. The dependent variable was the hourly waiting count in the ED, our operational measure of overcrowding. Seven regression models were developed to assess different predictors across various timestamps. **Results**: Weather conditions were significantly correlated with increased ED waiting count in the Baseline Model, while federal holidays and weekends were consistently correlated with reduced waiting counts. Boarding count was positively correlated with ED waiting count when concurrent, but boarding counts 3 h and 6 h before showed significant negative correlations. Hospital census showed a negative correlation in the Baseline Model but shifted to a positive effect in other models, reflecting its time-dependent influence on ED operations. Football games 12 h before significantly correlated with increased waiting counts, while games 12 and 24 h after had no significant effects. **Discussion**: While existing research typically focuses on limited variables and narrow timeframes, the temporal relationships between operational and non-operational factors affecting ED overcrowding remain understudied, particularly the delayed impacts of external events and environmental conditions. **Conclusions**: This study emphasizes the importance of incorporating both operational and non-operational factors to understand ED patient flow. Identifying robust predictors such as weather conditions, federal holidays, boarding count, and hospital census can inform dynamic resource allocation strategies to mitigate ED overcrowding effectively.

## 1. Background and Significance

Emergency Departments (EDs) face a widespread crisis of overcrowding, with over 90% routinely operating beyond capacity during peak periods [1]. This systemic challenge, where emergency service demand exceeds available patient care resources, correlates with increased mortality. Studies demonstrate that when ED occupancy surpasses average levels, in-hospital mortality rates increase by approximately 3.1%, rising to 5.4% during peak overcrowding [2]. The problem manifests through extended wait times, hallway boarding, and overwhelmed staff, creating cascading effects throughout healthcare systems that compromise both patient safety and care quality [3,4].

Various approaches to quantifying ED overcrowding have evolved over the past two decades, ranging from simple metrics to sophisticated scoring systems. The ED Occupancy Rate calculates the ratio of total patients to treatment beds [5], while Length of Stay (LOS) for admitted patients serves as another key indicator of ED backup [6]. More sophisticated tools include the Emergency Department Work Index (EDWIN) [7], which incorporates five variables: number of ED patients in each triage category, patient triage category, number of treatment spaces in the ED, i.e., bays or beds, number of attending physicians currently on duty, and number of admitted patients in the ED, i.e., holds or inactive patients. The National Emergency Department Overcrowding Scale (NEDOCS) [8] incorporates seven variables, including total patients, bed capacity, ventilator use, and boarding times [9]. The Community Emergency Department Overcrowding Scale (CEDOCS) [10] builds upon NEDOCS by incorporating community hospital-specific variables, while the Severely Overcrowded, Overcrowded, and Not-overcrowded Estimation Tool (SONET) [11] introduces composite indices for total patient load and waiting room status. Despite these innovations, achieving adequate standardization across healthcare settings remains challenging.

ED overcrowding significantly impacts patient outcomes, correlating with increased mortality, longer hospital stays, and higher healthcare costs [12]. Asplin et al.’s framework [13] provides a comprehensive structure for understanding this crisis through three interconnected components: input, throughput, and output. Input factors encompass both operational elements that hospitals can directly manage and non-operational influences such as environmental conditions, temporal patterns, and community events [14]. Moreover, limited access to primary care and specialty services drives increased ED utilization [15,16], while inadequate mental health resources direct crisis patients to EDs [17]. Additionally, insurance status significantly affects ED usage, particularly among populations with limited healthcare access alternatives [18,19].

Throughput factors reflect internal ED efficiency, where workflow organization, staffing levels, and ancillary service delays create potential bottlenecks in patient processing [20,21]. The boarding of inpatients due to limited bed capacity significantly impacts throughput by consuming resources needed for new patients [22]. Output factors are related to patient disposition, where organizational inefficiencies and delayed discharges create ripple effects throughout the hospital [23]. While existing research typically focuses on limited variables and narrow timeframes, the temporal relationships between factors remain understudied [5,12].

Our study addresses these gaps by developing a comprehensive set of models analyzing factors affecting ED waiting room counts, a key overcrowding indicator [24]. Following Asplin’s framework [13,14], we examine three variable categories. Input factors include temporal elements (hour, day, month, weekend status) and external influences (sporting events, holidays, weather). Throughput factors encompass treatment count and real-time waiting room metrics, while output factors include boarding count and hospital census. Specifically, we test the hypothesis that ED overcrowding is influenced by both immediate operational factors and time-lagged effects of non-operational factors, with these relationships varying across different temporal windows.

To capture temporal dynamics, we analyze these variables across multiple time lags, examining both immediate values and rolling averages. We incorporate lagged values of variables from 3 h before to 24 h after reference points. These temporal dimensions are crucial because while operational factors typically have immediate effects that hospitals can actively manage, non-operational factors often have delayed or sustained impacts requiring different management strategies. Further, our focus on external influences like weather conditions and sporting events is particularly significant, as prior research has shown that weather patterns can significantly affect ED utilization [25], while mass gatherings like sporting events can create sudden surges in ED demand through both direct injuries and indirect effects on community behavior [26,27].

## 2. Materials and Methods

### 2.1. Study Design and Setting

This was a retrospective observational study using secondary data analysis conducted at an academic medical center in the southeastern United States.

### 2.2. Data Sources

Four main data sources were used to investigate factors that impact ED overcrowding: ED tracking data, inpatient data, weather information, and significant event dates. The ED tracking and inpatient data originated from an academic medical center located in the southeastern United States. We start accessing on 10 November 2024. Weather data was obtained from the OpenWeatherAPI [28], using data collected from a station near the hospital. Significant event dates include federal holidays and game times for a major local college football team in the nearby city. All four datasets cover the period from 1 January 2019 to 1 July 2023, and were merged on an hourly basis to create a comprehensive dataset for analysis. The ED tracking dataset includes detailed records of patient visits, allowing tracking of patient movement within the ED. The dataset contains 161,477 unique patient IDs and 308,196 unique visit IDs. The dataset captures timestamps for ED arrival and discharge, as well as room codes that record when patients enter and exit various areas within the ED. Additionally, it includes an event column that records actions such as discharge, bed requests, triage, or physician exams throughout a patient’s visit. The inpatient dataset includes records for all patients admitted to the hospital, containing 209,505 unique patient IDs. For each admission, the dataset includes admit and discharge timestamps. The weather dataset includes hourly environmental data collected using OpenWeatherAPI’s history bulk. It contains numerical variables such as temperature, humidity, and wind speed, along with categorical weather statuses like clear or cloudy skies, rain, mist, thunderstorm, snow, drizzle, haze, fog, and smoke.

Finally, the significant events dataset includes categorical variables for major events that could impact hospital activity, specifically local football games and federal holidays. The selection of football games as the sporting event variable was based on attendance magnitude and the average stadium capacity of 60,000 attendees. In Alabama, college football games draw six to seven times more attendance than basketball. Due to this attendance differential, football games represented a suitable proxy measure for studying the impact of large-scale sporting events [29,30]. Football game times were sourced from the team’s official website [31], with an average of 13 games occurring each year. Federal holiday dates were obtained from the U.S. government website [32], covering 10 different holidays each year.

### 2.3. Operationalization of Variables and Data Preprocessing

We carried out several preprocessing steps to prepare the dataset for analysis. Each variable specified in Table 1 was aggregated on an hourly basis, with each row representing data for a specific one-hour interval. ED overcrowding refers to the overall imbalance between ED demand and resources. We operationalized ED overcrowding through the waiting count, our dependent variable. The following variables were derived from the ED tracking dataset:Waiting count (dependent variable): Represents the total number of patients who arrived at the ED at a specific hour but did not start treatment.Treatment count: Represents the total number of patients who are being treated in the ED at a specific hour.Boarding count: Represents the total number of patients whose treatments in the ED have been completed and who received admission orders from their emergency physician but remain in the ED while waiting for available beds in the inpatient units at any time within the hourly interval.The hospital census was derived from inpatient data, representing the total number of patients that are in inpatient units at any time within the hourly interval. Temperature, humidity, wind speed, and weather status were extracted from the weather dataset. Temperature and humidity were used to calculate the heat index using the heat index equation developed by Rothfusz [33]. The Football Game and Federal Holiday variables were sourced from the “significant events” dataset. These preprocessed data were then merged based on the hourly format, resulting in a comprehensive dataset for analysis. All the variables were structured to be on an hourly basis. A descriptive analysis of the final dataset is provided in Table 1.

We applied data-cleaning steps to enhance the quality of the data. The data from the COVID-19 period (from 3 January 2020 to 5 January 2021) was excluded, as hospital operations during this time were significantly different. Additionally, entries of patients who spent more than 9 h in the waiting room, accounting for 1.89% of the entire dataset, were removed based on the recommendation of our clinical collaborators. The weather status feature was also simplified by grouping conditions into five categories: clear, clouds, rain, thunderstorm, and others (including fog, haze, snow, and smoke). Each weather status is represented as a binary variable: a value of 1 indicates that the specific weather condition is present in the row, while a value of 0 indicates it is not. Additional variables were created from some of the original variables listed in Table 1 by using their values 3, 6, 12, and 24 h before and after the current time of the dependent variable (see Appendix A). This allows for capturing the temporal impact of those variables on waiting count. For instance, the thunderstorm variables 3 and 6 h before allowing for an analysis of how these can be used to explain the effect of time-shifted thunderstorm statuses on the current value of the dependent variable. Lastly, rolling means, which calculate the average of a variable over a specific time window, were applied to hospital census, treatment count, and boarding to assess the impact of these variables’ longer-term trends on waiting count.

### 2.4. Outcomes

The outcome variable in this study is waiting count, representing the ED overcrowding. The waiting count is the total number of patients in the waiting room at a specific one-hour interval. Patients were included in an hourly count if their waiting room arrival was before the end of the interval and their departure was after its beginning. The total count for each hourly interval was determined by counting all patients who met these criteria. For instance, to calculate the total number of patients at the 1:00 PM to 2:00 PM interval, any patient who arrives at the ED before 2:00 PM and departs anytime at or after 1:00 PM was counted for the interval. In another instance, a patient who arrives at 12:50 PM and departs at 2:10 PM was counted in the intervals 12–1 PM, 1–2 PM, and 2–3 PM.

### 2.5. Analysis

The primary analysis consisted of multivariate linear regression models to quantify the effects of operational and non-operational variables on ED waiting counts. Preliminary univariate analyses were conducted solely to guide variable selection and understand distributions before the main regression analysis. The statistical analyses were conducted in two steps. First, a preliminary analysis was performed to examine how individual independent variables influence the dependent variable (i.e., the waiting count). The primary objective of this analysis is to understand the distribution and impact of each variable on the dependent variable, providing a foundational understanding before moving to multivariate approaches. In the second step, multivariate linear regression was conducted to analyze the effects of the independent variables on the waiting count. This analysis aims to quantify the impact of various independent variables, including football game events, weather, and downstream ED metrics such as patient treatment and boarding count, on the number of waiting patients.

### 2.6. Preliminary Analysis

We started this analysis by inspecting the distributions of the numerical independent variables using histograms. Then, we explored the relationships between these variables, such as hospital census and treatment count, with the dependent variable using t-tests. This analysis enabled us to identify trends, correlations, and potential outliers. For categorical variables, such as weather status, federal holidays, and football game events, we used t-tests to assess the statistical significance of the difference in the waiting count between different categories. Additionally, we used confidence interval plots to illustrate the difference in the dependent variable across these categories, highlighting the direction and magnitude of the difference. The preliminary analysis provides initial insights into the variables that may contribute to the waiting count, setting the stage for subsequent multivariate analysis. Square root transformation was applied to the dependent variable to achieve normally distributed residuals.

### 2.7. Multivariate Regression Analysis

To further investigate the relationship between the dependent variable (hourly waiting count) and independent variables (downstream metrics and external factors such as weather), seven multivariate linear regression models were built. We selected linear regression over machine learning approaches to ensure interpretability of coefficients for operational decision-making and to quantify the specific contribution of each predictor variable. The models were designed to investigate the effects of independent variables at different timestamps on the dependent variable. The model design included a baseline set of variables, then extended to models that incorporate different timeframes before and after the current value of the independent variables to assess the impact of temporal changes on waiting count values, ensuring the robustness of the proposed regression analysis. In the Baseline Model, concurrent independent variables such as football game actual time, hospital census, treatment count, and weather statuses like clear, clouds, rain, and thunderstorm were included. This served as a foundation for understanding how these independent variables relate to the dependent variable.

To further enhance the robustness of the analysis and capture temporal variations, we introduced additional models that incorporate variables reflecting different time frames: Model 2 (3 hours Before) builds on the baseline by retaining key variables such as month, day of month, hour, day of week, heat index, and wind speed, while including additional features such as weather conditions (clear, clouds, rain, thunderstorm), hospital census data, and treatment and boarding counts to three hours before the time of the target event. In other words, Model 2 (3 hours Before) explains the effects of those variables *t* − 3, on the dependent variable at *t*. Additionally, football game time was shifted to 12 h before the target event.

Model 3 (6 hours Before) retains the same dependent variables as Model 2 but uses 6 h early values (*t* − 6 values), except for the football game variable, which uses 24 h before the target event. This design ensured that any significant trends manifesting over a longer time period were captured, hence enhancing the model’s robustness. Model 4 (12 hours After) and Model 5 (24 hours After) include similar sets of variables to the base model, except that Model 4 considers a 12 h timeframe for the football game, while Model 5 considers 24 h after. The inclusion of variables reflecting post-event dynamics provides valuable insights into how these variables influence patient flow beyond immediate effects.

Models 6 and 7 introduce rolling averages for key variables such as treatment count and hospital census data. Models 6 and 7 include treatment and boarding counts calculated over 6 h window. Model 6 includes hospital census calculated over 12 h, while model 7 includes hospital census calculated over 24 h. These rolling averages capture cumulative effects over time, allowing the model to consider not only instantaneous impacts but also the influence of trends over extended periods.

Incorporating variables across different timeframes results in a comprehensive and robust regression analysis. This design ensures that the model can generalize effectively to different scenarios, considering the dynamics of patient flow in the waiting room under varying temporal and environmental conditions.

### 2.8. Ethics Statement

This study involved the secondary analysis of limited patient data obtained from an academic medical center in the southeastern United States. The research protocol was reviewed and approved by the Institutional Review Board (IRB) at the University of Alabama at Birmingham (UAB), under approval number IRB-300011584. The IRB conducted an expedited review under Categories 5 and 7 and granted a waiver of informed consent and HIPAA authorization. Although the dataset included elements such as visit date and time, all analyses were conducted in compliance with IRB-approved data security and privacy protocols.

## 3. Results

### 3.1. Preliminary Analysis Results

We analyzed the effects of categorical variables on ED waiting counts using *t*-tests (α = 0.05, assuming equal variances), with each binary variable coded as “Yes” for the presence of the event (e.g., Federal Holiday) and “No” for its absence. Figure 1 displays confidence intervals and *p*-values for mean differences between categories. Football game timing was associated with notable variations in waiting counts: games held 24 h before (Figure 1B) were linked to increased waiting counts, whereas those 12 h after (Figure 1E) corresponded to decreased counts, while games at the current time (Figure 1A), 12 h before (Figure 1C), and 24 h after (Figure 1D) showed no significant effect. Federal holidays (Figure 1F) were consistently associated with lower waiting counts. Weather conditions also influenced ED crowding—clear weather (Figure 1G) corresponded to lower waiting counts, whereas cloudy (Figure 1H), rainy (Figure 1J), and thunderstorm (Figure 1K) conditions were associated with higher counts. The “Other” weather category, which includes drizzle, fog, haze, snow, and smoke (Figure 1I), also showed increased waiting counts, though these events occurred infrequently.

### 3.2. Multivariate Regression Results

The multivariate regression results presented in Table 2 provide insights into how various factors influence the total waiting count and how these effects evolve with the timing of certain independent variables, such as football games and weather conditions (See Appendix A in the Appendix A). This underscores the robustness of the model design. To fulfill the normality assumption of linear regression (See Appendix A in the Appendix A), we applied square root transformation to the dependent variable. The regression coefficients were used to interpret the impact of the independent variables on the square root of waiting count, meaning that for each unit of change in a certain independent variable, the square root of the waiting count changes by the corresponding coefficient.

In the Baseline Model, weather variables showed positive associations with the square root of the waiting count, with thunderstorms exerting the largest effect among all weather categories. Federal holidays and weekends significantly reduced waiting counts, indicating lower ED crowding during these periods. Operational factors behaved as expected: boarding count was positively related to waiting count, whereas hospital census showed a significant negative relationship. Temporal variables (hour, month, and day of the month) captured normal cyclical and seasonal variations in patient flow.

In Model 2, which examined variables measured three hours before the target time, weather conditions showed no significant influence on current waiting counts. Federal holidays and weekends remained significant predictors of lower ED crowding, though with slightly smaller effects than in the Baseline Model. Hospital census three hours earlier became positively associated with waiting count, while boarding count retained a significant but opposite relationship compared to the concurrent model. Notably, football games 12 h before showed a significant positive association with waiting counts, suggesting that crowding tends to rise in the hours leading up to major events.

In Model 3, which assessed variables measured six hours before the target time, weather effects remained statistically insignificant despite slightly stronger negative trends compared with Model 2. Federal holidays and weekends continued to show significant reductions in waiting counts, consistent with earlier models. Wind speed six hours before retained a significant positive association. Boarding count six hours earlier maintained a negative relationship similar to the three-hour lag model, while the effect of hospital census increased in magnitude. The football game 24 h before variable showed a small, nonsignificant increase in ED waiting counts, reinforcing that pre-event rather than post-event timing is more relevant to ED crowding.

Models 4 and 5 evaluated the effects of football games occurring 12 and 24 h after the index time. Across both models, federal holidays and weekends consistently retained significant negative associations with waiting counts, confirming their stable influence on reduced ED activity. Weather conditions maintained positive effects similar to those in the Baseline Model. In contrast, football games after the event showed no significant relationship with waiting counts, unlike the pre-event period where a 12 h lead was significant. This pattern suggests that increases in ED crowding are driven primarily by pre-event behaviors rather than post-event factors.

Models 6 and 7 incorporated rolling averages of hospital census and boarding count to capture cumulative operational effects. In Model 6, the rolling mean of boarding count became negative, while hospital census turned positive compared with their directions in the Baseline Model, suggesting that prolonged high hospital occupancy contributes to increased ED waiting counts over time. In Model 7, both rolling averages remained negative but with smaller, nonsignificant effects. Comparing the two models indicates that shorter averaging windows (e.g., 12 h) better capture the delayed impact of inpatient load on ED crowding than longer windows (24 h), reflecting the time-sensitive nature of hospital throughput dynamics.

Model performance metrics (R^2^ and adjusted R^2^) indicate consistent explanatory strength across specifications. The Baseline Model accounted for about 28% of the variance in the square root of waiting counts, with modest improvement in the 3 h and 6 h lag models. Models incorporating rolling averages (Models 6 and 7) demonstrated the best fit, explaining nearly 39% of the variance. These results suggest that including lagged and cumulative operational variables meaningfully enhances the models’ ability to capture temporal patterns in ED crowding.

## 4. Discussion

Our findings contribute to the growing body of literature on ED overcrowding by highlighting the temporal dynamics of both operational and non-operational factors. Extending the conceptual framework proposed by Asplin et al. [13,14], this study quantifies specific time-lagged effects that were previously hypothesized but not empirically validated. Unlike prior research that focused primarily on concurrent relationships, our temporal analysis demonstrates that the influence of environmental, community, and hospital-level factors on ED crowding depends strongly on timing, offering a more nuanced understanding of the mechanisms driving congestion. Weather conditions emerged as significant contributors, with thunderstorms and wind speed showing the strongest associations with higher waiting counts, likely due to increased accidents or exacerbations of chronic conditions during adverse weather. These findings align with Dring et al. [34], who reported a long-term rise in heat-related ED visits, and with studies linking extreme weather to increased mortality [35]. However, our observation that clear weather corresponds to reduced waiting counts diverges from the literature focusing on extreme weather effects, suggesting behavioral rather than purely environmental drivers. The absence of lagged weather effects indicates that meteorological influences on ED utilization are immediate, likely reflecting rapid behavioral or physiological responses. The consistent significance of wind speed across all models represents a novel finding, reinforcing recent evidence of heightened ED use during adverse conditions [36].

The timing of football games also influenced waiting counts, though this relationship was shaped more by behavioral and operational dynamics than event magnitude. While simple comparisons suggested changes before and after games, multivariate models clarified that only games occurring approximately 12 h before were significantly associated with higher waiting counts, implying that pre-event gatherings and travel contribute more to ED congestion than the games themselves. This temporally specific finding refines previous work by Jerrard and Hughes et al. [37,38], who reported reduced ED visits during game hours, and is consistent with Antonwiak et al. [39], who observed modest decreases during regional and Monday night games. Operational factors such as hospital census and boarding counts demonstrated complex, time-dependent relationships. High real-time boarding counts correlated with greater ED congestion, while earlier boarding activity showed weaker or negative associations, likely reflecting patient-flow cycles. Similarly, hospital census showed short-term negative but lagged positive effects, suggesting that sustained inpatient occupancy eventually amplifies ED crowding. Rolling-average models confirmed that prolonged strain on inpatient capacity contributes cumulatively to waiting-room pressure, consistent with the capacity-threshold frameworks described by Janke et al. [40] and Kelen et al. [41]. Federal holidays and weekends consistently corresponded with reduced waiting counts, likely reflecting lower urgent-care demand and altered hospital operations, contrasting with studies showing higher total visits due to non-urgent cases [42,43]. Temporal factors such as hour and month captured predictable diurnal and seasonal variations, reinforcing the cyclical nature of ED demand. Collectively, these findings demonstrate that ED overcrowding arises from the timing and interplay of multiple determinants rather than from isolated factors. While several variables showed statistically significant but modest coefficients, their cumulative and time-sensitive effects have meaningful implications for forecasting, staffing, and operational planning. Future research should examine these interactions to support predictive tools for real-time ED management and improve system resilience.

These findings carry important implications in the context of climate change and rising global temperatures. As heat waves, severe storms, and extreme weather events become more frequent and intense, the temporal patterns we identified, particularly the immediate effects of thunderstorms and wind speed, may represent early indicators of escalating pressure on ED systems worldwide. Climate-driven increases in acute health events, combined with strain on hospital capacity, are likely to amplify overcrowding risks and resource challenges in the coming decades. Integrating weather and climate projections into ED forecasting models may therefore be essential for improving resilience and preparedness in emergency care systems globally [44,45].

From an operational perspective, these small effect sizes suggest that ED managers should not focus on single variables in isolation but rather consider them as part of a comprehensive management strategy. For instance, while a 0.002 change in boarding count coefficient may seem negligible for a single hour, when multiplied across thousands of patient encounters annually, these effects accumulate to operationally meaningful impacts. When combined with weather events and temporal patterns, the annual cumulative effects become highly relevant for budgeting, staffing decisions, and resource allocation strategies. These findings inform ED resource allocation, suggesting that additional staff should be deployed immediately during severe weather, rather than in advance, and also recommend staffing 12 h before football games.

## 5. Limitations

This study, while offering valuable insights into ED overcrowding, had multiple limitations. First, the analysis was based on data from a single southeastern U.S. academic medical center. This might limit the generalizability of findings to other regions or healthcare systems due to varying patient demographics, workflows, policies and regulations, or environmental conditions. Multi-center validation studies are essential to confirm whether these temporal relationships and effect sizes are consistent across different ED settings and patient populations. The regression models rely on linear relationships and temporal shifts, which may not capture complex, non-linear interactions among variables. Advanced machine learning approaches could provide deeper insights into these dynamics. Finally, preprocessing steps such as excluding patients waiting more than nine hours may inadvertently omit extreme cases that could provide valuable insights into severe overcrowding scenarios.

## 6. Conclusions

The regression results highlight the multifaceted and time-dependent nature of ED overcrowding. Weather conditions, particularly thunderstorms and wind speed, are significantly associated with waiting count, while football games influence waiting counts primarily 12 h before the event. Hospital census and boarding count exhibit complex, lagged effects, with cumulative metrics providing additional insights into their sustained impacts. Federal holidays and weekends consistently reduce waiting count, while temporal variables capture predictable variations in patient flow. It is also noteworthy that some variables, despite being statistically significant, have relatively small coefficients, suggesting limited practical impact when considered individually. These findings highlight the importance of incorporating temporal and cumulative metrics into ED management strategies and considering the combined effects of multiple factors to dynamically allocate resources and mitigate ED overcrowding effectively.

## Figures and Tables

**Figure 1 healthcare-13-02577-f001:**
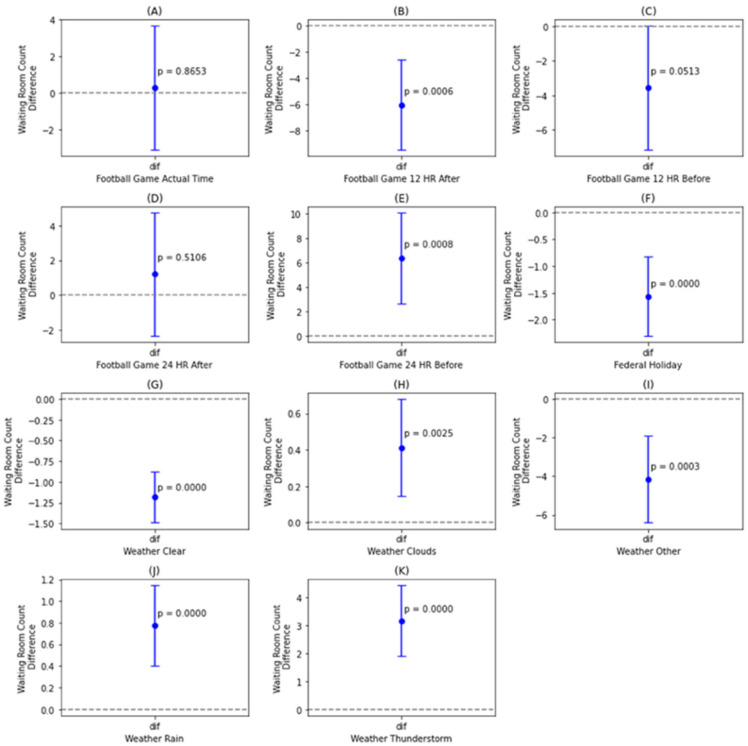
Confidence interval plots for ED waiting count stratified by: football game occurrence and proximity (**A**–**E**); federal holiday status (**F**); and weather conditions-clear (**G**), cloudy (**H**), other (**I**), rain (**J**), and thunderstorm (**K**).

**Table 1 healthcare-13-02577-t001:** Summary of the Dataset.

Feature	Ranges for Date/Time Variables, Average ± Standard Deviation (Range) for Numerical Variables, % for Categorical Variables, and Count for Event Variables
Month of Year	1–12 Months
Day of Month	1–31 Days
Day of Week	1–7 Days
Hour of Day	1–24 Hours
Waiting Count (Dependent Variable)	18.23 ± 9.86 (0–59)
Treatment Count	67.37 ± 23.52 (9–139)
Boarding Count	45.72 ± 29.27 (3–121)
Hospital Census	792.29 ± 71.66 (584–1017)
Heat Index	86.2 ± 24.9
Wind Speed	2.62 ± 2.05 m/s (0–15.4)
Weather Status	
Clouds	60.10%
Clear	22.78%
Rain	15.55%
Thunderstorm	1.25%
Others	0.33%
Football Game	43 Games (across the entire date range)
Federal Holidays	37 Days (across the entire date range)

**Table 2 healthcare-13-02577-t002:** Regression coefficients for predictors of ED waiting counts across temporal models.

	Baseline Model	Model 2: 3 Hours Before	Model 3: 6 Hours Before	Model 4: 12 Hours After	Model 5: 24 Hours After	Model 6: Rolling Averages	Model 7: Rolling Averages 2
Football Game Actual Time	0.104					0.140	0.134
Hospital Census	−0.003 (***)			−0.003 (***)	−0.003 (***)		
Boarding Count	0.002 (***)			0.002 (***)	0.002 (***)		
Federal Holiday	−0.636 (***)	−0.448 (***)	−0.365 (***)	−0.635 (***)	−0.635 (***)	−0.409 (***)	−0.509 (***)
Weather Clear	1.058 (***)			1.059 (***)	1.058 (***)	0.512 (***)	0.757 (***)
Weather Clouds	1.064 (***)			1.064 (***)	1.064 (***)	0.486 (***)	0.740 (***)
Weather Rain	1.004 (***)			1.004 (***)	1.003 (***)	0.449 (***)	0.691 (***)
Weather Thunderstorm	1.182 (***)			1.181 (***)	1.181 (***)	0.605 (***)	0.877 (***)
Heat Index	−0.003 (***)	−0.002 (***)	−0.001 (***)	−0.003 (***)	−0.003 (***)	−0.002 (***)	−0.002 (***)
Wind Speed	0.042 (***)			0.042 (***)	0.042 (***)	0.038 (***)	0.040 (***)
Weather Others	0.574 (***)			0.575 (***)	0.575 (***)	−0.012	0.227 (**)
Month of year	0.004 (*)	0.001	−0.002	0.004 (*)	0.004 (*)	−0.002	0.001
Day Of Month	−0.005 (***)	−0.004 (***)	−0.004 (***)	−0.005 (***)	−0.005 (***)	−0.004 (***)	−0.005 (***)
Hour of day	0.072 (***)	0.073 (***)	0.067 (***)	0.072 (***)	0.072 (***)	0.074 (***)	0.076 (***)
Weekend	−0.800 (***)	−0.624 (***)	−0.554 (***)	−0.799 (***)	−0.800 (***)	−0.598 (***)	−0.677 (***)
Weather Clear 3 Hours Before		−0.245					
Weather Clouds 3 Hours Before		−0.143					
Weather Rain 3 Hours Before		−0.206					
Weather Thunderstorm 3 Hours Before		−0.188					
Weather Other Hours Before		−0.411					
Wind Speed 3 Hours Before		0.049 (***)					
Boarding Count 3 Hours Before		−0.002 (***)					
Hospital Census 3 Hours Before		0.001 (***)					
Football Game 12 Hours Before		0.305 (*)					
Football Game 12 Hours After				−0.122			
Weather Clear 6 Hours Before			−0.736				
Weather Clouds 6 Hours Before			−0.566				
Weather Rain 6 Hours Before			−0.646				
Weather Thunderstorm 6 Hours Before			−0.549				
Weather Other Hours Before			−0.549				
Wind Speed 6 Hours Before			0.034 (***)				
Boarding Count 6 Hours Before			−0.004 (***)				
Hospital Census 6 Hours Before			0.003 (***)				
Football Game 24 Hours Before			0.207				
Football Game 24 Hours After					0.146		
Rolling Mean Boarding Count Window Size 6						−0.003 (***)	−0.001 (*)
Rolling Mean Hospital Census Window Size 12						0.002 (***)	
Rolling Mean Hospital Census Window Size 24							−0.000

* *p* < 0.05, ** *p* < 0.01, *** *p* < 0.001.

## Data Availability

Data supporting the findings of this study are available from the authors upon reasonable request and subject to approval by the Institutional Review Board (IRB) of the University of Alabama at Birmingham (UAB).

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
