# Peer review of "Assessing the Impact of External and Internal Factors on Emergency Department Overcrowding"

_healthcare, 2025, doi:10.3390/healthcare13202577_

Round 1
Reviewer 1 Report
Comments and Suggestions for Authors
This is an interesting article examining emergency department boarding integrating causal factors, such as weather or football games. The topic itself is quite interesting, but there are hundreds of papers examining this topic and I'm not sure about the level of significance of this study over the others. I would encourage the author team to try to generalize significance more: for instance, you use football games but what about the impact of baseball or basketball? Do we expect similar results? Why weren't other types of events included, or what is the rationale behind only football games? The basic premise underlying your findings is that greater volume of people in the commmunity equates to higher ED wait times (more people at games, more people in ED; less people due to holiday, lower volumes). I wish the results could be a little more interesting than that.
Please look at the section under Data Sources, and remove the second instance of the sentence "We start accessing the data on 10/11/2024." You use that exact sentence twice in the same paragraph.
Reviewer 2 Report
Comments and Suggestions for Authors
Dear Authors,
Thank you for the opportunity to read and revise your manuscript.
Your research addresses a timely and relevant issue like ED overcrowding and external variables (weather conditions, federal holidays, football games) as potential determinants. Your findings, particularly on weather effects and the impact of football events, represent novel contributions.
The four-year analysis is a strength, while the single-center study may limit the generalizability of your findings.
I hope my comments could be useful to improve your work. Please, see comments in the attached PDF file.
All the best
_____________________________________________
Keywords: to improve indexing and discoverability, I recommend replacing or adding the PubMed MeSH term “Emergency Department”.
Count/crowding/boarding: I found some inconsistencies in the use of these terms, potentially confusing your findings (and the readers). Crowding refers to the overall imbalance between ED demand and resources, while boarding is a specific output factor contributing to crowding. In the manuscript, “waiting count” seems used as the operational definition of crowding, and “boarding count” is modeled as a predictor. I suggest clarifying these definitions early in the methods section and maintaining consistent terminology throughout.
Study design and setting: A clear definition of the study type is missing. In accordance with current academic standards, the manuscript would benefit from a clearer description of the study type: this appears to be a retrospective observational study using secondary data analysis. I suggest reporting the study design (both Abstract and Methods sections) to enhance methodological clarity and indexing.
The name of the hospital where the research was conducted is missing (you only mention “an academic medical center in the southeastern United States”). For clarity and reproducibility, the institution should be reported.
The section on Operationalization of Variables and Data Preprocessing is well structured and detailed, providing clear definitions of dependent and independent variables. This level of methodological clarity strengthens reproducibility. However, the definitions are currently reported in long sentences, making it difficult to follow the text flow (especially for readers unfamiliar with ED terminology). To improve readability, you may edit these as a bullet-point list.
Hospital census: I suggest making this definition clearer for readers unfamiliar with ED/hospital terminology. You may consider a short, plain explanation (e.g., “Hospital census represents the total number of inpatients occupying hospital beds at a given hour, serving as a proxy for overall hospital occupancy”).
I suggest adding some details for “football game” (if available, or add this as a limitation) , like stadium capacity (fixed number ) and expected attendance per game (varying). This may help quantify the number of potential ED patients.
Results: this section is rich in data but at times overly detailed and redundant. Several numerical values are described extensively in the text even though they are already included in Table 2. To improve readability, I suggest streamlining the section, focusing the text on the most relevant and novel findings (e.g., thunderstorms, hospital census lagged effects, football games), leaving the detailed numbers in the tables.
Given the density of data and the number of tables, I recommend adding a graphical abstract. This may help highlight your findings, making the manuscript more attractive and readable. This may improve your effectiveness in communicating the main findings to a broader audience.
Discussion: the discussion could be strengthened by addressing the implications of rising global temperatures and climate change, as the increasing frequency of heat waves, severe storms, and extreme weather events may raise the burden on EDs worldwide. To enhance the public health relevance of your research, I suggest speculating on how your findings might be interpreted in the context of long-term climate trends, potentially amplifying ED overcrowding challenges. Useful references: https://doi.org/10.1016/S2542-5196(25)00075-0 - https://jphe.amegroups.org/article/view/10131/pdf
The discussion section is informative but excessively long, as it frequently repeats numerical results already presented in Table 2 and the Results. I recommend focusing this section to emphasize interpretation, mechanisms, and implications, removing unnecessary numerical values.
The current discussion is comprehensive but lacks a clear take-home message to link your findings to concrete operational decisions. You should add a short “Implications for practice” paragraph (at the end of the discussion).

Reviewer 3 Report
Comments and Suggestions for Authors
The study is about a really important topic for emergency medicine, and it looks at the many reasons why EDs get overcrowded. I found the paper really informative and enjoyable to read. This manuscript provides a clear and engaging overview of emergency department (ED) overcrowding, including factors such as weather, football games, and holidays. The large dataset and the use of multiple regression models make the findings pretty robust. The study addresses a significant clinical and operational issue and provides some practical tips for managing EDs. The paper is really well written, informative and a pleasure to read.
I have some recommendations to improve the paper.
You can make the results clearer by adding simple figures or tables to show the impact of the weather and football games.
Give a quick chat about the clinical or operational importance of variables with small effect sizes, even if they're statistically significant.
Just add a little more to the bit about the single-centre design and say how important it is to check it in other healthcare systems too.
Can you just say why we used linear regression and why we didn't go for something more complex, like machine learning?
Edit the discussion to cut out some of the repetitive statements and make it flow better. The effects of weather (like thunderstorms, wind speed and clear weather), the 12-hour pre-game football effect, the lagged relationships of hospital census and boarding, and the consistent impact of federal holidays/weekends are all described in a similar way in both the Results and Discussion sections. We could cut down on these repetitions by putting the findings together into one clear paragraph, which would make it easier to read and keep us on track.
Round 2
Reviewer 2 Report
Comments and Suggestions for Authors
I have no additional requirements.
Thank you for this revised version.